# Effect of Process Parameters on Energy Consumption, Physical, and Mechanical Properties of Fused Deposition Modeling

**DOI:** 10.3390/polym13152406

**Published:** 2021-07-22

**Authors:** Emmanuel U. Enemuoh, Stefan Duginski, Connor Feyen, Venkata G. Menta

**Affiliations:** 1Mechanical & Industrial Engineering, University of Minnesota Duluth, Duluth, MN 55812, USA; eenemuoh@d.umn.edu (E.U.E.); dugin012@umn.edu (S.D.); 2Dept. of Material Science, University of Pennsylvania, Philadelphia, PA 19104, USA; cfeyen@seas.upenn.edu

**Keywords:** fused deposition modeling, additive manufacturing, Taguchi orthogonal array, energy consumption, tensile strength

## Abstract

The application of the fused deposition modeling (FDM) additive manufacturing process has increased in the production of functional parts across all industries. FDM is also being introduced for industrial tooling and fixture applications due to its capabilities in building free-form and complex shapes that are otherwise challenging to manufacture by conventional methods. However, there is not yet a comprehensive understanding of how the FDM process parameters impact the mechanical behavior of engineered products, energy consumption, and other physical properties for different material stocks. Acquiring this information is quite a complex task, given the large variety of possible combinations of materials–additive manufacturing machines–slicing software process parameters. In this study, the knowledge gap is filled by using the Taguchi L_27_ orthogonal array design of experiments to evaluate the impact of five notable FDM process parameters: infill density, infill pattern, layer thickness, print speed, and shell thickness on energy consumption, production time, part weight, dimensional accuracy, hardness, and tensile strength. Signal-to-noise (S/N) ratio analysis and analysis of variance (ANOVA) were performed on the experimental data to quantify the parameters’ main effects on the responses and establish an optimal combination for the FDM process. The novelty of this work is the simultaneous evaluation of the effects of the FDM process parameters on the quality performances because most studies have considered one or two of the performances alone. The study opens an opportunity for multiobjective function optimization of the FDM process that can be used to effectively minimize resource consumption and production time while maximizing the mechanical and physical characteristics to fit the design requirements of FDM-manufactured products.

## 1. Introduction

Additive manufacturing (AM) can be defined as the process of making products by joining engineering materials layer upon layer using computer-aided design (CAD) data [1]. AM is finding its place in a wide scope of engineering and biological industries due to its ability to produce free-form complex parts in small- to medium-sized batches, making it competitive in the world economy and meeting the demands of the growing dynamic market in terms of the increasing demand of customized and personalized products, reduced lead times, and sustainability [2]. AM has the capability of printing functional parts using a wide range of materials such as steel, ceramic, and polymers. Amongst the different AM systems, which include fused deposition modeling (FDM), stereolithography (SLA), selective laser sintering (SLS), direct metal laser sintering (DMLS), direct metal deposition (DMD), inkjet modeling (IJM), laminated object manufacturing (LOM), etc., that are available commercially for the manufacturing of layered parts, FDM has been found to be the most promising and robust AM technique that can produce complex and intricate parts in clean, safe, and reasonable time requirements [3]. FDM has widened its scope of applications from prototypes for visual conceptual models and the form–fit test to the production of functional parts such as drilling grids in the aerospace industry [4]. However, FDM continues to face challenges in its usage to produce functional parts due to its different shortcomings with mechanical properties, dimensional accuracy, surface finish, and energy footprint. Dey and Yodo conducted extensive reviews of the state-of-the-art literature on the influence of FDM parameters on part qualities and the existing work on FDM process parameter optimization [5]. They concluded from their review that polylactic acid (PLA) and acrylonitrile butadiene styrene (ABS) were the most widely used materials for producing functional parts with FDM. They also identified the shortcomings and challenges of existing works and evaluated opportunities to work in the FDM additive manufacturing field and suggested directions for future research in this field. They concluded that process parameters such as infill pattern, print speed, shell width, or extrusion temperature are less analyzed compared to layer thickness, build orientation, raster width, or raster orientation [5]. The factors affecting FDM processes were classified into process-related and product-related control factors, and there is limited research in the optimization of multiple FDM parts’ quality characteristics [5]. Chacon et al. characterized the effect of build orientation, layer thickness, and feed rate on the mechanical performance of PLA. They showed that on-edge and flat orientation possessed higher strength and stiffness than the upright orientation. They reported that the mechanical properties of FDM prints in the upright orientation will increase with an increase in layer thickness and decrease with an increase in feed rate. For the on-edge orientation, they reported slight variations in mechanical properties with changes in layer thickness and feed rate, except for the case of low layer thickness [6]. Reverte et al. studied the effect of short carbon fiber on the mechanical and geometric properties of FDM polylactic acid composites. The addition of carbon fibers effectively improved the mechanical properties of PLA-CF composites as compared to the neat PLA. The flat PLA-CF samples showed an average increase in tensile performance of 47.1% for the tensile strength and 179.9% for the tensile stiffness in comparison to the neat PLA. They reported an average increase in average flexural strength and stiffness of 89.75% and 230.95%, respectively, in comparison to the neat PLA. Furthermore, PLA-CF samples depicted the best interlaminar shear strength (ILSS) performance. The use of short carbon fiber as reinforcement did not affect the dimensional accuracy of the PLA-CF samples and improved the surface roughness in the flat and on-edge orientations [7]. Chacon et al. determined that the effect of nozzle diameter on the mechanical and geometric quality of the FDM of fiber-reinforced PLA has statistical significance. The mechanical performance and surface roughness were increased with large nozzle diameters with a reduction in manufacturing costs. On the other hand, a small nozzle diameter produced higher geometric accuracy. The flatness error and surface roughness were not significantly influenced by the FDM nozzle diameter [8]. 

On the aspect of energy consumption, the United States Energy Information Administration (EIA) named the manufacturing sector as a significant source of energy consumption, and the driving accessories in the manufacturing sector alone consume around 1.35 × 10^19^ J each year, generating about 521 MT CO_2_ Eq. amount of carbon emission [9]. Since 2010, the United States has been recording the largest increase in energy consumption every year in both absolute and percentage terms. Hence, optimizing current processes, adapting new techniques, and developing new technologies is paramount to limit the increase in energy consumption and reduce carbon emissions [10]. The FDM is one such technology with the potential of improving energy efficiency in the manufacturing industry. Weissman and Gupta found that the energy consumption of FDM is highly dependent on the volume and geometry of the products [11]. Peng reported that the printing speed and material flow rate have a small effect on the particulate emission rate, while heating the print bed and maintaining the bed temperature consume the most energy [12]. He also reported that the carbon footprint increases as the shape of the part becomes more complex. Mognol et al. performed tests under various parameter level combinations: part orientations and positions in AM systems. They concluded that minimizing manufacturing time is critical to reducing energy optimization for all systems [13]. In comparing the surface roughness and energy consumption, Peng and Yan found that the layer thickness is the most influential factor that generated opposite effects, followed by infill ratio and printing speed [12]. The authors also found that a higher printing speed can effectively reduce energy consumption and maintain good surface roughness [12]. Griffiths et al. used a design of experiments approach for part optimization with the consideration of scrap weight, part weight, energy consumption, and production time [14]. According to Enemuoh et al., about 95.5% of the energy consumed during the FDM of PLA was seen during the actual layer-by-layer part building, implying that process parameters such as infill density, layer thickness, and print speed can be used to significantly improve the energy footprint of FDM [15]. It was shown that through the optimization of machine build parameters, a desired response is possible, and compromises between output responses such as scrap weight and production time can be identified. 

Most of the parametric studies reported in the literature evaluated either the mechanical performance or energy consumption of FDM. However, it is important for the end user to know the effects the process parameters have on both mechanical and physical performance and energy consumption of FDM. In order to close research gaps identified in the literature, Taguchi’s L_27_ orthogonal array design of experiments was used to evaluate and quantify the main effects of the five major distinguished FDM process parameters: infill density, infill pattern, layer thickness, print speed, and shell thickness on multiple quality characteristics: energy consumption, production time, dimensional changes, part weight, hardness, and tensile strength. The FDM process parameters are among the parameters identified as needing more attention by Dey and Yodo [5], except the layer thickness. Their extensive review article comprised about one hundred papers on the influence of FDM parameters on part qualities and the existing work on FDM process parameter optimization [5]. They stated that the effects of many FDM process parameters, including extrusion temperature, number of shells, infill pattern, and raster width, on dimensional accuracy are still unknown. The layer thickness parameter is included because the FDM process is fundamentally based on layer-by-layer part manufacturing. The tensile strength of the FDM part was most studied, and it has been established that the build orientation was the most significant parameter. The tensile strength was maximum at a zero-degree (or flat/on-edge) build orientation because the direction of filament fiber extrusion is parallel to the direction of the applied load. In this study, the build orientation will not be varied. Figure 1 is the cause and effect diagram to show the different factors that would influence the different FDM quality characteristics. Signal-to-noise (S/N) ratio analysis and analysis of variance (ANOVA) will be performed on the experimental data to quantify the parameters’ main effects on the responses and establish optimal combinations for the multiple FDM quality characteristics. The models established from this study can be used to effectively minimize resource consumption and production time while maximizing the mechanical properties characteristics to fit the design requirements of FDM-manufactured products.

## 2. Materials and Methods

### 2.1. Material

Polylactic acid (PLA) is a thermoplastic material made from natural lactic acid from natural resources, contrary to other petroleum-based thermoplastics. The properties of PLA are comparable to other plastics, and as a result, there is a considerable desire to introduce it into the plastic market as a competitive material. It is an environmentally friendly thermoplastic characterized by compostability, biodegradability, and biocompatibility. Its melting temperature is within the range of 180–230 °C [10]. It is available in natural (translucent white) or in many bright colors, solid or half transparent, and the printed objects have a beautiful smooth surface. PLA can be used to manufacture flexible joints, belts, tires, etc. It can also be used for medical applications, for example orthopedic devices, replacing temporary titanium screws and sutures [16]. In this project, PLA with polycarbonate fortification (PLA PRO) is employed to take advantage of its better tolerances and properties. The PLA PRO used has a tensile strength of 37 MPa.

### 2.2. Design of Experiment

Taguchi’s robust L_27_ orthogonal array analytical procedure was used to investigate the five major FDM process parameters and their alternate levels, as shown in Table 1. Taguchi’s orthogonal array design of experiments was developed to investigate how different process parameters affect the mean and variance of a process performance characteristic that defines how well the process is performing [17]. The parameters to study and their alternate levels are organized using orthogonal arrays. Unlike in full factorial design of experiments, where all combinations of factor levels are tested, the Taguchi method tests pairs of combinations. As the name suggests, the columns of this array are mutually orthogonal. Here, orthogonality is interpreted in the combinatorial sense; that is, for any pair of columns, all combinations of factor levels occur, each an equal number of times. This is called the balancing property, and it implies orthogonality [17]. This property reduces the complexity of full factorial experiments to twenty-seven simple and effective experiments. In this case, a full factorial design of 53 = 125 experiments is reduced to 27 experiments. Taguchi analysis was performed using Minitab 16 software to understand the influence of FDM parameters on the measured quality characteristics. The Taguchi method is best used when there is an intermediate number of variables and with few interactions between the variables such as in this study. 

The FDM’s layer thickness, infill density, infill pattern, print speed, and shell thickness comprise a Taguchi L_27_ orthogonal array [17]. The array defines the twenty-seven test conditions needed to analyze an FDM process parametrically. The levels selected for the experiments were chosen so as not to obscure the influence of any parameter on the measured energy consumption, physical, and mechanical properties. Additionally, each of the twenty-seven array experiments was repeated two times to evaluate the variability associated with a given test condition and to reduce experimental errors. The layer thickness of the FDM part is the height of the deposited layers of the filament along the *z*-axis, which is generally the vertical axis of the FDM machine. The layer thickness is typically less than the diameter of the FDM extruder nozzle and would depend on the nozzle diameter. It is generally recommended that the layer thickness not exceed 80% of the nozzle diameter. Because a 0.4 mm diameter nozzle was used in this study, a maximum value of 0.3 mm, a minimum value of 0.1 mm, and a midway value of 0.2 mm were selected as the layer thickness levels. The infill density is the percentage of infill volume with filament material. It would commonly consist of the invisible inner structure of the FDM part covered by the outer layer(s) and has different shapes, sizes, and patterns. The infill density is expected to have an influence on the energy consumption, strength, hardness, production time, and weight of the FDM build parts. The three wide spread levels were chosen to not obscure the influence of each level on the FDM quality characteristics. On the other hand, different infill patterns such as triangle, gyroid, and cubic can be used during FDM part building. The different infill patterns will produce different levels of strength and durable internal structure of the FDM part. In the current study, cubic, gyroid, and triangle infill patterns are considered. A cubic or honeycomb pattern is the most commonly used 3D printing infill pattern and generally provides relatively uniform strength in all directions. Gyroid is known for providing high strength values at lower weights. Triangle patterns are typically used for parts with few connections between walls such as thin and rectangular geometries. The print speed is the distance traveled by the FDM extruder per unit time along the XY plane while extruding filament material. The print speed is measured in mm/s and would influence the production time, energy consumption, and other quality characteristics of the FDM process. The recommended speed for most FDM printers is 60 mm/s. Increasing the print speed to very high values will result in poor adhesion between layers due to insufficient cooling time. Reducing the print speed to very low values results in part deformation due to the high nozzle dwell times over the plastic. Hence, 40 mm/s and 80 mm/s were selected as lower and upper bounds in the current work. Shells are the outer perimeters in each layer, and shell thickness also refers to the number of shells in the print. While 0.8 mm is commonly used for shell thickness, a multiple of nozzle diameter is recommended. In the current work, 0.4 mm and 1.2 mm were selected as the lower and upper levels, while 0.8 mm was selected for the middle level. The experimental responses or quality characteristics in this design include energy consumption, production time, part weight, dimensional accuracy, hardness, and tensile strength are shown in Figure 2. 

### 2.3. Experimental Setup

The FDM process was conducted with an in-house FDM Ultimaker S5 three-dimensional (3D) printer with a heated build plate maintained at 60 °C. The layer resolution can range from 20 to 300 microns depending on the nozzle diameter. The environmental conditions of air pressure, air temperature, and humidity were recorded and within 1% variability. A raster orientation of 45 degrees was used for all the samples. The Ultimaker Cura was used to slice and prepare the CAD model for printing. During each experimental run, an energy meter was used to measure the energy consumption. The sample dog bone shape was based on the ASTM D638 Type IV standard, which is used for the tensile testing of plastic materials [1]. The dimensions of the dog bone shape sample are shown in Figure 3, and the 3D printed sample is shown in Figure 4.

Each experimental run was conducted two times, making it a total of 54 samples to allow for residual analysis. The samples were carefully labeled and stored in a vacuum-tight storage bin for further evaluation of their dimensional tolerances, followed by weight measurements. The Shore D hardness measurements of the samples were taken using a PTC Instruments ASTM Type D Model 307L durometer with four replications per sample, and average values were used for analysis. After the nondestructive quality characteristics evaluations of the samples were completed, the final destructive tensile tests were conducted on the 54 samples to evaluate their tensile strength properties. The tensile test was conducted with an Applied Technical Services (ATS) machine with a 44.48 KN maximum load cell. The load cell range for the test and test speed was 2.224 KN and 6.35 mm/min, respectively. Controlled by an Adam Bradley controller, a preload of roughly 440 N was applied to each specimen before the test. 

## 3. Results

The responses obtained from the experiments were analyzed using a graphical representation of the means effects and the analysis of variance (ANOVA) on the FDM quality characteristics. Interaction effects between the control factors were ignored as they were minimal. The responses analysis helped in identifying those process parameters that have the greatest impact on FDM process variability and its level of performance. To determine this, signal-to-noise (S/N) ratio analysis was used as shown in Equations (1) and (2). The transformation method was used to convert the measured responses into an S/N ratio. Proposed by Taguchi, S/N ratios are performance measures that optimize a process. S/N ratio analysis also provides a sensitivity measurement of the quality characteristics of a process at various levels of both controllable and uncontrollable factors (or noise). Thus, the optimum process design is achieved when the S/N ratio is maximized [17]. In other words, it is the process condition at which the variability, resulting from the uncontrollable factors, is minimized or maximized depending on the measured response. Equation (1) is used to evaluate the S/N ratio for “smaller is better” responses in the study, which includes energy consumption, production time, and part weight. Equation (2) is used to evaluate the “larger is better” responses, which include the FDM part tensile strength and hardness.
(1)SN=−10Log10(1n(∑i=1nYi2))
(2)SN=−10Log10(1n(∑i=1n1Yi2))

*Y_i_* is the measured response for the *i*th test part, and n represents the number of test parts for an experimental run. The largest or minimum *S*/*N* ratios indicate optimal factor levels that minimize the noise sensitivity.

The *p*-values and *F*-values from the ANOVA of all the FDM quality characteristics from the experiments are summarized in Table 2. The low *F*-value shows a case of low variability, where the control factor’s between-level means are close together relative to the variability within each level. The high *F*-value shows a case where the variability of the control factor’s means is large relative to the within-group variability. A high *F*-value is needed to reject the null hypothesis that the level means are equal, meaning that the control factor has statistical significance on the quality characteristics. Statistical significance of the control factor exists at *p* < 0.05. A level of *p* = 0.05 indicates a 5% risk of concluding that an association exists when there is no actual association.

### 3.1. FDM Part Energy Consumption 

The S/N means effect of the FDM parameters on energy consumption is illustrated in Figure 5a. The responses represent changes due to the control factor level changes. The *p*-values in Table 2 show that all the control factors have a significant effect on energy consumption, except for the shell thickness. The amount of change in energy consumption obtained in Table 3 should coincide with the statistical significance obtained in the ANOVA results. The means of energy consumption in Table 3 show that layer thickness has the highest effect on the energy consumed during FDM production, followed by the print speed, infill density, infill pattern, and shell thickness. This provides a reliable basis for selecting optimal process parameters. The S/N ratios in Figure 5a suggest that the optimized factor levels that generated the minimum energy consumption were estimated to be A3/B1/C1/D3/E1. These optimal factor levels indicate a layer thickness of 0.3 mm, an infill density of 20%, a triangle infill pattern, a print speed of 80 mm/s, and a shell thickness of 0.4 mm.

### 3.2. FDM Part Production Time

The S/N means effect of the FDM parameters on production time is illustrated in Figure 5b. The change in production time represents changes due to the investigated control factor levels. The observed change in production time coincides with the statistical significance obtained in its ANOVA. The effect of the control factors on the FDM production time had a similar ranking as in the case of energy consumption, as shown in Figure 5b and Table 4. Hence, optimizing the energy consumption results in the direct optimization of production time, and both S/N ratios are computed using Equation (1). The S/N ratios in Figure 5b suggest that the optimized factor levels that generated the minimum production time were estimated to be A3/B1/C1/D3/E1. These optimal factor levels indicate a layer thickness of 0.3 mm, an infill density of 20%, a triangle infill pattern, a print speed of 80 mm/s, and a shell thickness of 0.4 mm.

### 3.3. FDM Part Weight

The changes in the FDM part weight represent changes due to the investigated control factor level. It is a “smaller is better” kind of quality characteristics, and its S/N ratio is maximized using Equation (1). The S/N means effects of the control factors on the FDM part weight were evaluated and are illustrated in Figure 5c. The number of response changes obtained in Table 5 coincides with the statistical significance obtained in the ANOVA, which shows that all the control factors have statistical significance on the weight of the FDM part. The S/N ratios in Figure 5c suggest that the optimized factor levels that generated the minimum part weight were estimated to be A3/B1/C1/D2/E1. These optimal factor levels indicate a layer thickness of 0.3 mm, an infill density of 20%, a triangle infill pattern, a print speed of 60 mm/s, and a shell thickness of 0.4 mm.

### 3.4. FDM Part Dimensional Accuracy

The changes in the FDM part dimensional accuracy represent changes in the width dimension of the sample test region due to the investigated control factors levels. The smallest changes are desired, and its S/N ratio is evaluated using Equation (1) (smaller is better). The S/N means effects of the control factors on the dimensional changes are illustrated in Figure 5d, and the number of response changes obtained in Table 6 coincides with the statistical significance obtained in the ANOVA results. The *p*-values in Table 2 show that only shell thickness and layer thickness have a statistically significant effect on the FDM part’s dimensional changes. The S/N ratios in Figure 5d suggest that the optimized factor levels that generated the minimum FDM dimensional changes were estimated to be A1/B1/C2/D2/E1. These optimal factor levels indicate a layer thickness of 0.1 mm, an infill density of 60%, a gyroid infill pattern, a print speed of 60 mm/s, and a shell thickness of 0.4 mm.

### 3.5. FDM Part Hardness

The FDM part hardness quality characteristic was evaluated using the larger is better criteria, and its S/N ratio is maximized using Equation (2). The S/N mean effect of the control factors on the part hardness was evaluated and is illustrated in Figure 5e. The hardness represents changes due to the investigated control factor level changes. The number of response changes obtained in Table 7 coincides with the statistical significance of the obtained ANOVA results. The *F-* and *p*-values in Table 2 show that only layer thickness has a significant effect on the hardness of the FDM part. The means effects of the control factors on the part hardness and their rankings are shown in Table 7, with the layer thickness having a dominant effect at the lowest level. The S/N ratios in Figure 5e suggest that the optimized factor levels that generated the maximum part hardness were estimated to be A1/B2/C1/D2/E1 or E2. These optimal factor levels indicate a layer thickness of 0.1 mm, an infill density of 60%, a triangle infill pattern, a print speed of 60 mm/s, and a shell thickness of 0.4 mm or 0.6 mm.

### 3.6. FDM Part Tensile Strength

The FDM part tensile strength quality characteristics is a larger-is-best response type, and its S/N ratio is maximized using Equation (2). The means effect of the control factors on the FDM part tensile strength was evaluated and is illustrated in Figure 5f. The changes in the parts’ tensile strength represent changes due to the investigated control factor level changes. The number of response changes obtained coincide with the statistical significance obtained by the ANOVA. The means effects of the control factors on the part tensile strength and their ranking are shown in Table 8. The S/N ratios in Figure 5f suggest that the optimized factor levels that generated the maximum FDM part tensile strength were estimated to be A2/B3/C3/D1/E3. These optimal factor levels indicate a layer thickness of 0.2 mm, an infill density of 100%, a cubic infill pattern, a print speed of 40 mm/s, and a shell thickness of 1.2 mm.

## 4. Discussion

The signal-to-noise ratio, analysis of variance (ANOVA), and analysis of means techniques were used to successfully predict the relative significance and ranking of the FDM process control factors, experimental errors, and models’ term coefficients. They provided the percentage contribution of each FDM control factor, thus providing a better feel for the relative effect of the different factors and their levels on the FDM quality characteristics considered as presented in Section 3. The summary of control factors S/N means and their optimum levels for the FDM quality characteristics are shown in Table 9. It is remarkable that FDM layer thickness, print speed, and infill density dominated in contributing the highest effects on the six quality characteristics investigated. The FDM layer thickness ranked highest for energy consumption, production time, and hardness. Peng and Yang [19] had a similar result on energy consumption. The infill density ranked highest for the part weight and tensile strength, while the print speed ranked second for energy consumption and production time. The control factors influence the mechanical properties by affecting the extrusion rate and morphology of the printed parts. When the printed FDM part retains incompletely fused tiny pores and cracks due to less than optimum printing conditions, the parts’ strength may decrease due to stress concentrations at the areas where pores exist; energy consumption and production time are also affected.

The behavior of the FDM control factors is common to the energy consumption and production time. Therefore, the optimization of the control factors to minimize energy consumption will result in minimum production time that will lead to minimum manufacturing costs and minimal environmental impact. The FDM layer thickness defines the step height (*Z*-axis). To obtain parts with minimum energy and minimum production time, the layer height should be set high (0.3 mm). However, to obtain the best FDM hardness and tensile strength properties, it is necessary to choose a small layer height (0.1 mm), which would create a tighter fusion between layers; however, it would take a longer time to print. The ideal solution is to find a balance between the production time, energy consumption, and mechanical properties needed for the design intention of the FDM part. The first layer height is usually defined as a percentage of the normal layer height.

The FDM Infill density defines the infill density as 0% being hollow and 100% being solid. Naturally, the solid part will exhibit superior mechanical and physical properties than the hollow or semi-hollow FDM parts. This is the reason infill density played a big role in the six quality characteristics of the FDM part. Decisions about the optimal levels of the FDM part should consider the design goals, production time, and energy consumption. For FDM prototypes, it is usual to use an infill percentage between 30 and 40%, and for functional parts that require full rigidity, it should be around 70–80% at minimum.

The FDM shell thickness represented the number of layers on the outside of an FDM part. Observing the summary presented in Table 9, shell thickness had the second highest S/N rankings of the FDM hardness and tensile properties. The FDM part tensile strength can be improved by increasing the shell thickness. A similar result was reported for the compression strength of FDM PLA [2]. This allows for a slightly more robust FDM print without having to increase the amount of material used for infill. If the FDM part is to be finished by sanding or chemical smoothing, increasing the shell thickness is often necessary, as postprocessing methods reduce the thickness of the surface of the FDM part. It is noteworthy that any increase in the number of shells would also increase the amount of time and material required to print the FDM part, therefore increasing overall production time, energy consumption, and cost. The rule of thumb is to design shells to be a multiple of the nozzle diameter to prevent voids from being formed in the FDM part [20].

The FDM print speed contributed the second highest S/N effect on energy consumption and production time. It also contributed to the tensile strength of the FDM parts due to the molecular chain entanglement between layers. During the FDM process, the temperature of the polymer melt decreases after it is extruded from the nozzle; hence, increasing the print speed maintains the temperature of the polymer layer at a desired level for a longer time. However, too high a print speed may lead to print imperfections and inferior part properties, while an optimal print speed allows a molecular chain to bond at the interfaces with superior mechanical property [21].

The infill pattern has a relatively small influence on the five quality characteristics investigated, with their Delta ranging from 0.06 to 0.53 dB. The triangle infill pattern produced the highest S/N for energy consumption, production time, weight, and hardness of the FDM part. The cubic pattern had the highest S/N for the tensile strength. The cubic infill pattern is closest to the rectangular infill that can achieve a 100% dense part because it consists of a grid of parallel and perpendicular extrusions.

### Verification of Experiments

The optimal level settings listed in Table 9 for each quality characteristic within the region of this study were used to predict the signal-to-noise ratio of the optimal quality characteristics, as shown in Equations (3) through (8). Because interaction effects were ignored, as discussed in Section 2, the equations yield the following mean S/N ratio of the quality characteristics: energy consumption = 22.04 dB; production time = −28.32 dB; part weight = −13.31 dB; dimensional changes = 8.48 dB; hardness = 38.16 dB; tensile strength = 35.23 dB.
$$Energy-\eta_{A3/B1/C1/D3/E1}=\eta_{m}+\left(\eta_{A3}-\eta_{m}\right)+\left(\eta_{B1}-\eta_{m}\right)+\left(\eta_{C1}-\eta_{m}\right)+\left(\eta_{D3}-\eta_{m}\right)+\left(\eta_{E1}-\eta_{m}\right)\;$$
(3)$$Energy-\eta_{A3/B1/C1/D3/E1}=\eta_{A3}+\eta_{B1}+\eta_{C1}+\eta_{D3}+\eta_{E1}-4\eta_{M}\;$$
(4)$$\;Prod.\;Time-\eta_{A3/B1/C1/D3/E1}=\eta_{A3}+\eta_{B1}+\eta_{C1}+\eta_{D3}+\eta_{E1}-4\eta_{M}\;$$
(5)$$Weight-\eta_{A3/B1/C1/D3/E1}=\eta_{A3}+\eta_{B1}+\eta_{C1}+\eta_{D2}+\eta_{E1}-4\eta_{M}\;$$
(6)$$Tolerance-\eta_{A3/B1/C1/D3/E1}=\eta_{A1}+\eta_{B2}+\eta_{C2}+\eta_{D2}+\eta_{E1}-4\eta_{M}\;$$
(7)$$Hardness-\eta_{A3/B1/C1/D3/E1}=\eta_{A1}+\eta_{B2}+\eta_{C1}+\eta_{D2}+\eta_{E1}-4\eta_{M}\;$$
(8)$$Strength-\eta_{A3/B1/C1/D3/E1}=\eta_{A2}+\eta_{B3}+\eta_{C3}+\eta_{D1}+\eta_{E3}-4\eta_{M}$$

*η_M_* is the mean *S*/*N* ratio for the experimental test runs, and *η_ij_* is the *S*/*N* ratio of control factor *i* at the optimal level *j* setting.

A new set of experiments with three replications was conducted to analyze and validate the established optimum levels of the FDM process for the different quality characteristics. The *S*/*N* evaluation of the quality characteristics was calculated, and the results indicate that the optimized control factor levels from the study adequately minimized the energy consumption, production time, part weight, and dimensional changes, while the hardness and tensile strength properties were adequately maximized as predicted. Figure 6 shows an example of each printed sample at the optimal level settings. 

The following equivalent average S/N responses were evaluated: energy consumption = 21 dB; production time = −29 dB; part weight = −13.6 dB; dimensional changes = 8.5 dB; hardness = 38.27 dB; tensile strength = 34.3 dB. Figure 7 shows that all the predicted quality characteristics of FDM parts are within a 5% margin of error with their respective verification experiments.

## 5. Conclusions

In this study, Taguchi’s L_27_ orthogonal array design of experiments was used to study and quantify the effects of major FDM control factors—layer thickness, infill density, infill pattern, and print speed—on the multiple quality characteristics of FDM parts simultaneously. Signal-to-noise ratio mean effect analysis and analysis of variance (ANOVA) were used to draw the following conclusions.

During the FDM of PLA material with Ultimaker S5, layer thickness (Delta = 0.1811 dB, Rank = 1) has the largest mean effect on energy consumption, followed by print speed (Delta = 0.0939 dB, Rank = 2) and then by infill density, infill pattern, and shell thickness. The *p*-values from the ANOVA show that the association between the energy consumption and the term’s coefficient of layer thickness, infill density, infill pattern, and print speed control factors in the model are statistically significant. The shell thickness does not have a significant association at an alpha level of 0.05.The ranking of the mean effect of the control factors on production time followed the same order as energy consumption. The layer thickness (Delta = 58.78 dB, Rank = 1) has the largest mean effect on production time, followed by print speed (Delta = 30.56 dB, Rank = 2) and then by infill density, infill pattern, and shell thickness. The *p*-values from the ANOVA show that the association between the production time and the term’s coefficient of layer thickness, infill density, infill pattern, print speed, and shell thickness in the model are statistically significant at an alpha level of 0.05.The FDM layer thickness (Delta = 3.74, Rank = 1) has the highest effect on the hardness of the FDM part. The *p*-values from the ANOVA show that FDM’s infill density, infill pattern, print speed, and shell thickness do not have a statistically significant association with FDM part hardness at an alpha level of 0.05.The FDM part weight has the highest mean effect from infill density (Delta = 2.089 dB, Rank = 1), followed by layer thickness (Delta = 0.297 dB, Rank = 2) and then by print speed, infill pattern, and shell thickness. The *p*-values from the ANOVA show that the association between the FDM part weight and the term’s coefficient of all the control factors are statistically significant.The infill density has the highest mean effect on tensile strength of the FDM part with a Delta of 15.7 dB. The shell thickness (Delta = 3.49 dB) has the second highest mean effect and is followed by layer thickness, print speed, and infill pattern. The print speed and infill pattern do not have a significant association with the tensile strength at an alpha level of 0.05.The shell thickness has the highest effect on the dimensional changes of the FDM part with a Delta of 0.145 dB. The layer thickness (Delta = 0.116 dB) followed the rank of the effect of control factors. The infill density, infill pattern, and print speed do not have a statistically significant association with the dimensional changes of the FDM part at an alpha level of 0.05.

## Figures and Tables

**Figure 1 polymers-13-02406-f001:**
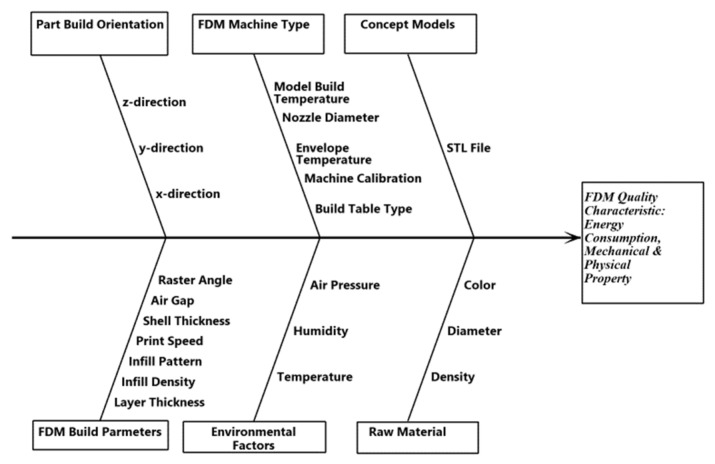
Cause effect diagram of the FDM quality characteristics.

**Figure 2 polymers-13-02406-f002:**
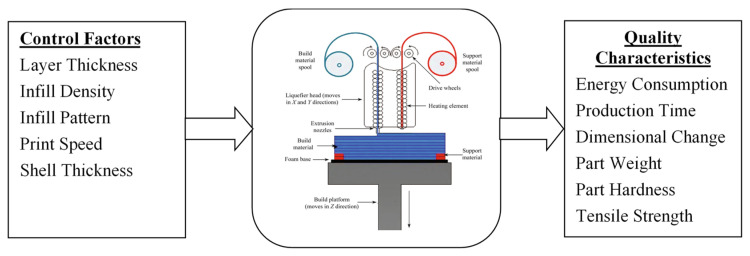
Framework for FDM part quality characteristics study, adopted from Mohamed et al. [18].

**Figure 3 polymers-13-02406-f003:**
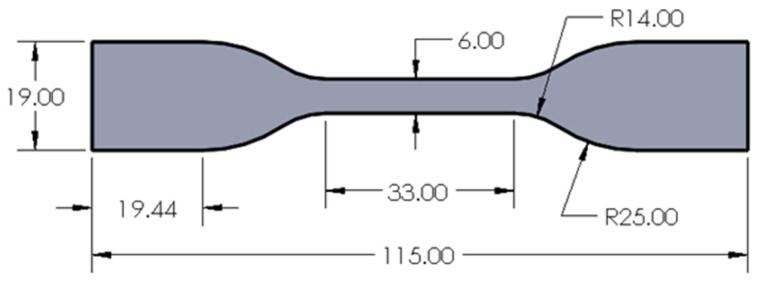
Geometric shape of the samples built. All dimensions are in mm.

**Figure 4 polymers-13-02406-f004:**
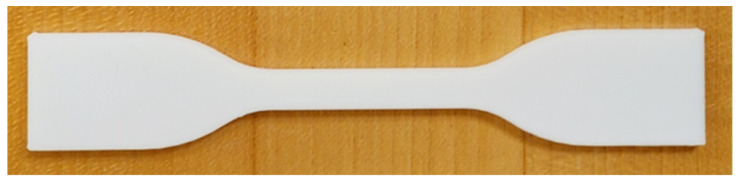
FDM PLA sample with a 100% infill density using Ultimaker S3.

**Figure 5 polymers-13-02406-f005:**
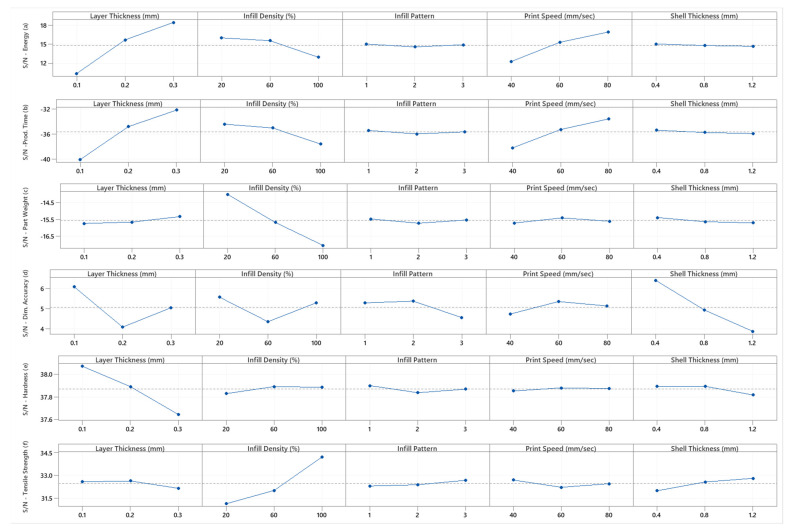
Mean of the S/N ratios of the FDM part quality characteristics (layer thickness-A; infill density-B; infill pattern-C; print speed-D; shell thickness-E).

**Figure 6 polymers-13-02406-f006:**
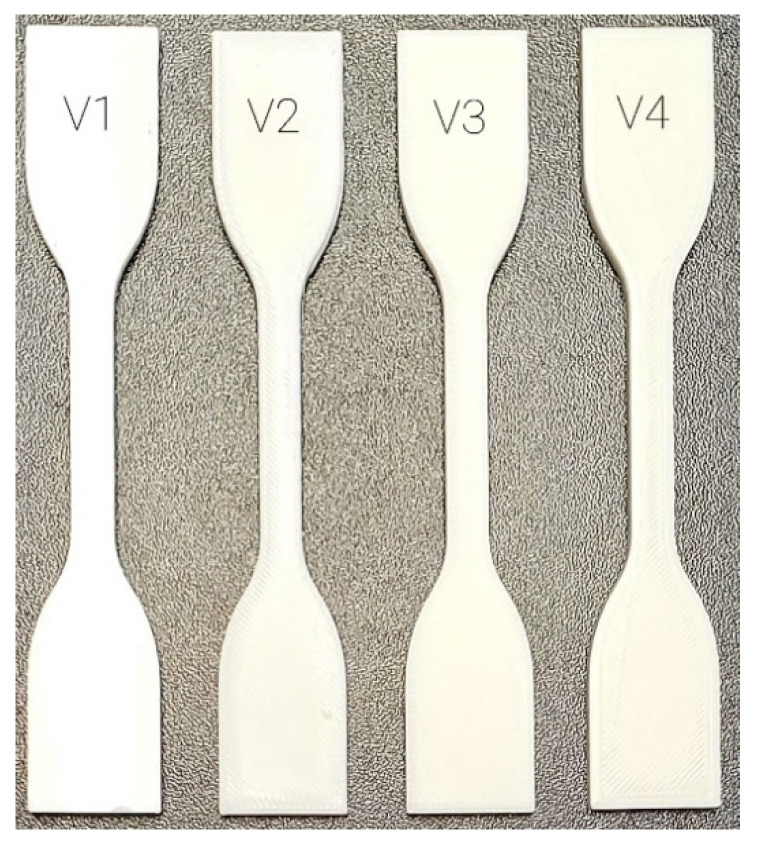
Samples produced with optimal levels.

**Figure 7 polymers-13-02406-f007:**
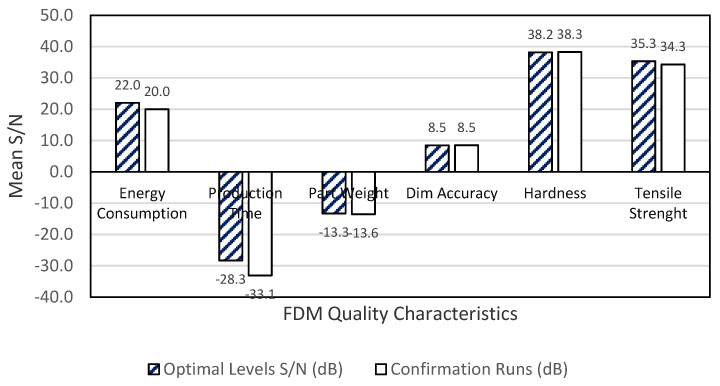
Comparison of Optimum levels with verification experiments.

**Table 1 polymers-13-02406-t001:** FDM control factors and their alternate levels.

FDM Control Factors	Control Factor Levels		
	1	2	3
Layer thickness (mm)	0.1	0.2	0.3
Infill density (%)	20	60	100
Infill pattern (degree)	Triangle	Gyroid	cubic
Print speed (mm/s)	40	60	80
Shell thickness (mm)	0.4	0.8	1.2

**Table 2 polymers-13-02406-t002:** Summary *F*- and *p*-values from the ANOVA of FDM quality characteristics.

Source	Energy Consumption (KWh)		Production Time (min)		Part Weight (g)		Dimensional Accuracy		Hardness (Shore E)		Tensile Strength (Pa)	
	F	P	F	P	F	P	F	P	F	P	F	P
Layer thickness (mm)	704.57	0.000	1254.79	0.000	19.32	0.000	4.99	0.021	43.69	0.000	3.66	0.049
Infill density (%)	81.50	0.000	158.15	0.000	802.44	0.000	2.43	0.120	1.15	0.342	120.70	0.000
Infill pattern	9.12	0.002	18.89	0.000	8.67	0.003	1.28	0.305	0.91	0.423	2.09	0.157
Print speed (mm/ss)	204.75	0.000	360.02	0.000	10.31	0.001	0.82	0.460	0.18	0.835	3.00	0.078
Shell thickness (mm)	3.18	0.069	6.61	0.008	7.29	0.006	7.73	0.004	1.80	0.196	5.88	0.012

**Table 3 polymers-13-02406-t003:** Response table for the S/N of energy consumption (Equation (1)).

Level	Layer Thickness (mm)	Infill Density (%)	Infill Pattern	Print Speed (mm/s)	Shell Thickness (mm)
1	10.39	15.99	15.01	12.29	15.03
2	15.68	15.55	14.58	15.30	14.80
3	18.43	12.95	14.91	16.91	14.67
Delta	8.04	3.05	0.42	4.62	0.36
Rank	1	3	4	2	5

**Table 4 polymers-13-02406-t004:** Response for the S/N of production time (Equation (1)).

Level	Layer Thickness (mm)	Infill Density (%)	Infill Pattern	Print Speed (mm/s)	Shell Thickness (mm)
1	−40.02	−34.43	−35.43	−38.17	−35.38
2	−34.81	−35.02	−35.96	−35.27	−35.75
3	−32.20	−37.58	−35.64	−33.59	−35.91
Delta	7.83	3.15	0.53	4.58	0.52
Rank	1	3	4	2	5

**Table 5 polymers-13-02406-t005:** Response table for the S/N of part weight (Equation (1)).

Level	Layer Thickness (mm)	Infill Density (%)	Infill Pattern	Print Speed (mm/s)	Shell Thickness (mm)
1	−15.73	−14.01	−15.47	−15.71	−15.38
2	−15.65	−15.67	−15.72	−15.40	−15.63
3	−15.33	−17.03	−15.53	−15.60	−15.70
Delta	0.41	3.02	0.25	0.30	0.31
Rank	2	1	5	4	3

**Table 6 polymers-13-02406-t006:** Response table for the S/N of part dimensional accuracy (Equation (1)).

Level	Layer Thickness (mm)	Infill Density (%)	Infill Pattern	Print Speed (mm/s)	Shell Thickness (mm)
1	6.080	5.568	5.287	4.735	6.398
2	4.092	4.357	5.369	5.352	4.939
3	5.047	5.293	4.563	5.132	3.881
Delta	1.988	1.211	0.806	0.617	2.517
Rank	2	3	4	5	1

**Table 7 polymers-13-02406-t007:** Response table for the means of FDM hardness (Equation (2)).

Level	Layer Thickness (mm)	Infill Density (%)	Infill Pattern	Print Speed (mm/s)	Shell Thickness (mm)
1	38.07	37.83	37.90	37.85	37.89
2	37.89	37.89	37.84	37.88	37.89
3	37.64	37.88	37.87	37.87	37.82
Delta	0.43	0.06	0.06	0.03	0.08
Rank	1	4	3	5	2

**Table 8 polymers-13-02406-t008:** Response table for the means of part tensile strength (Equation (2)).

Level	Layer Thickness (mm)	Infill Density (%)	Infill Pattern	Print Speed (mm/s)	Shell Thickness (mm)
1	32.59	31.15	32.31	32.71	32.00
2	32.65	32.02	32.40	32.23	32.58
3	32.16	34.23	32.69	32.47	32.82
Delta	0.49	3.09	0.38	0.48	0.82
Rank	3	1	5	4	2

**Table 9 polymers-13-02406-t009:** Summary of the S/N effect of control factors and their optimum levels (layer thickness—A; infill density—B; infill Pattern—C; print speed—D; shell thickness—E).

FDM Quality Characteristics	Highest Means S/N Effect	2nd Highest Means S/N Effect	3rd Highest Means S/N Effect	4th Highest Means S/N Effect	Optimum Levels Combination of Control Factors
FDM energy consumption	Layer thickness	Print speed	Infill density	Infill pattern	A3/B1/C1/D3/E1
FDM production time	Layer thickness	Print speed	Infill density	Infill pattern	A3/B1/C1/D3/E1
FDM part weight	Infill density	Layer thickness	Shell thickness	Print speed	A3/B1/C1/D2/E1
Dimensional tolerance	Shell thickness	Layer thickness	Infill density	Infill pattern	E1/A1/B2/C2/D2
FDM hardness	Layer thickness	Shell thickness	Infill pattern	Infill density	A1/B2/C1/D2/E1
FDM tensile strength	Infill density	Shell thickness	Layer thickness	Print speed	A2/B3/C3/D1/E3

## Data Availability

Not applicable.

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
