# Peer review of "Effect of Process Parameters on Energy Consumption, Physical, and Mechanical Properties of Fused Deposition Modeling"

_polymers, 2021, doi:10.3390/polym13152406_

Round 1
Reviewer 1 Report
This paper presents
- Some of the statements in this paper are not accurate. For example, the first sentence in the abstract significantly narrowed down what FDM can do. FDM methods have been used to 3D print both polymers and metals. Its applications are much broader.
- In table 1, the authors gave a list of factors and values for study. How these values are selected for each factor? Please explain.
- Please add units to table 2 for some of the parameters, such as production time.
- For polymer FDM 3D printing, some parameters that are not discussed in this paper have significant impacts on the printing quality, such as nozzle temperature, build plate temperature, leveling of the build plate, and initial distance between nozzle and build plate. Could the authors explain why they select the parameters reported in this paper instead of the others?
- Some of the reported results, such as tensile strength, depend on the printing direction. For example, the filament along the longitudinal direction of the gauge area of dogbone samples potentially has the highest tensile strength, and transverse to the longitudinal direction of the gauge area of dogbone has the lowest tensile strength. The authors did not provide filament printing directions for dogbone samples, therefore, it is impossible to evaluate the accuracy of the tensile test results reported in this paper.
- Please carefully proofread the draft and remove all comments before re-submission.
Author Response
Please see the section that addressed reviewer #1 comments/feedback.
Thank you very much!

Reviewer 2 Report
This manuscript represents the experimental analysis of the influence of different process parameter on energy efficiency and mechanical of Fused Filament Fabrication (FFF) technique. Layer thickness, infill densuty and pattern and feed rate have been studied on the manufacturing of 3D printed PLA specimens. The structure of the paper is well organized and the overall paper tells a logical story with a concrete conclusion. It is suitable for publication in Polymers after major revision.
1) Please explain in detail the novelty of this study and your scientific contribution.
2) Introduction and Reference sections should be improved with published works related to the analysis of the influence of process parameters on the mechanical and geometric performances of 3D printed thermoplastic structures, as for example:
[1] J.M. Chacón, M.A. Caminero, E. García-Plaza, P.J. Núñez, Additive manufacturing of PLA structures using fused deposition modelling: Effect of process parameters on mechanical properties and their optimal selection, Materials and Design 124 (2017) 143-157
[2] J.M. Reverte, M.A. Caminero, J.M. Chacón, E. García-Plaza, P.J. Núñez, J.P. Becar, Mechanical and Geometric Performance of PLA-Based Polymer Composites Processed by the Fused Filament Fabrication Additive Manufacturing Technique, Materials 13(8) (2020) 1924
[3] [ J.M. Chacón, M.Á. Caminero, P.J. Núñez, E. García-Plaza, J.P. Bécar, Effect of nozzle diameter on mechanical and geometric performance of 3D printed carbon fibre-reinforced composites manufactured by fused filament fabrication, Rapid Prototyping Journal 27(4) (2021) 769-784
3) Please include the basic mechanical properties of PLA thermoplastic material used in this study
4) Were the different manufactured plates analysed by ultrasonic or x-ray inspection in order to evaluate the manufacturing quality (inner defects, porosity, ...)
5) How many specimens were tested for each condition? Please include standard deviation of the experimental result
6) The graphical quality of the different figures depicted in the manuscript must be improved
Author Response
Please see the section that addressed reviewer #2 comments and feedback.
Thank you very much!

Reviewer 3 Report
Peer-Review polymers – 1288687
The manuscript entitled “Effect of Process Parameters on Energy Consumption, Physical, and Mechanical Properties of Fused Deposition Modeling” by Emmanuel U. Enemuoh et al. constitutes a well-designed study towards the optimization of Fused Deposition Modeling (FDM) process parameters and to achieve a better understanding of the effects of such parameters on the 3D printed construct properties.
The manuscript is well written, properly organized and fits within the scope of the journal polymers (ISSN 2073-4360). Nevertheless, some improvements must be made to enhance the overall manuscript quality before its consideration for publication.
Comments:
- (Page 2, 1.Introduction, line 52-54): The authors should be more descriptive about existing shortcomings and challenges as well as future directions. The statement in its current form is too general.
- In the Introduction, the authors should provide a deeper description of the Taguchi’s L27 orthogonal array design of experiments and discuss why this method is advantageous in relation to other methods.
- (Page 3, 2. Materials and Methods, Section 2.1. Material): The authors should use references to support the statements about the PLA material.
- (Page 3, Table 1): Please standardize the values of layer thickness values to the same format e.g., 0.X.
- (Page 5, 2. Materials and Methods, Figure 2): Please change/enhance the image quality to allow the reading of the text included in the middle box (3D-printer).
- (Page 5, 2. Materials and Methods, Figure 4 caption): Please specify 100% (of what) in the caption.
- (Page 5, 2. Materials and Methods, Section 2.3. Experimental Setup): The authors should specify and describe the conditions (e.g., load cell, strain rate, etc...) used in the tensile test.
- (Page 6, 3. Results, lines 201-205): This statement should be supported by a literature reference.
- (Page 6, 3. Results, line 222): Please standardize the values of p-value to the same format e.g., 0.X.
- (Page 7, 3. Results, Figure 5): The quality of Figure 5 should be considerably enhanced. Also, in the manuscript text the authors use Figure 5a, b, c (...), but these letters are not present in the image.
- (Pages 8/9, 3. Results, section 3.4. FDM Part Dimensional Accuracy, line 279 and Table 6): Looking to the Table 6 values it appears that the optimized factor levels would be A1/B1/C2/D2/E1 instead of A1/B2/C2/D2/E1 (stated in line 279). Please correct this.
- (Pages 9, 3. Results, Table 6): Please use the same font format of the other tables in the manuscript.
- (Pages 9, 3. Results, section 3.5. FDM Part Hardness, Table 7 and line 294): Based on the Table values, it may be more accurate to use A1/B2/C1/D2/E1 or E2 or something equivalent.
- (Pages 12, 4. Discussion, Figure 7): Please enhance considerably the quality of the graph and do not overlap the numeric values.
- (Pages 13, 5. Conclusions): Please standardize the delta values presented in the Conclusions section as some have units and others not.
Minor issues
- (Page 1, Abstract, lines 16,17): There is a missing comma “,” and “and” in the phrase. Please correct this.
- (Page 1, Abstract, line 18): It is better the plural form in (...)”Taguchi L27 Orthogonal Array design of experiments” (...).
- (Page 2, 1.Introduction, lines 71, 72): Please erase the repetition of the word “heating”.
- (Page 3, 2. Materials and Methods, Section 2.2. Design of Experiment): Please replace “influencing parameters” by “understand the influence of FDM parameters on the measured quality characteristics.”
- (Pages 10, 4. Discussion, line 314): Please remove the incorrect repetition of the word “used”.
- (Pages 12, 4. Discussion, line 387): Please correct the mistakes in the phrase to make it clearer.
- (Pages 12, 4. Discussion, line 398): Remove the misplaced/incorrectly used “and” before “dimensional changes”.
Author Response
Please the section in the attached cover letter that addressed the comments and feedback from reviewer #3.
Thank you very much!
Round 2
Reviewer 1 Report
The authors have addressed to all the comments given by the reviewer. The reviewer suggests publishing this paper in the current form.
Reviewer 2 Report
The paper has been improved with the answers to the Reviewers' comments. It is suitable for publication in present form.